# Position: Lifelong In-Context Learning Requires Parametric Forms of Attention

## Abstract

Lifelong continual learning remains an obstacle on the path to human-like intelligence. Modern transformers show sparks of intelligence with in-context learning. The quadratic nature of attention, however, prohibits transformers from performing this process on arbitrarily long sequences. In this work, we argue that extending in-context learning to lifelong settings is a practical solution for continual learning in AI agents. In particular, we argue that *parametric forms of attention* are needed to understand a lifetime of context with a fixed hardware budget. These attention mechanisms learn the relationship between keys and their associated values at test-time with parametric regression. Our generalization of parametric approaches (linear attention, state-space models, fast weight programmers, and test-time training layers) contrasts with nonparametric counterparts like softmax attention. They replace the ever-growing key-value cache with an online-trainable neural network, maintaining a constant memory footprint. We highlight how parametric forms of attention currently fall short of lifelong learning due to limited memory capacity or costly online updates. To address these issues, we pose a set of open questions with novel insights to guide the field toward long-horizon agents.

## 1. Introduction

Modern AI systems are trained *offline* on vast but finite datasets. When new capabilities are desired, the training recipe is extended (additional data, mid-training stages, etc.), and the model is retrained. This paradigm has been effective for short-horizon applications like chatbots, yet acting over longer horizons remains elusive.

[1]Anonymous Institution, Anonymous City, Anonymous Region, Anonymous Country. Correspondence to: Anonymous Author <anon.email@domain.com>.

Preliminary work. Under review by the International Conference on Machine Learning (ICML). Do not distribute.

Deployment becomes the dominant source of novel information as systems move toward agents that run for months or years, such as autonomous research agents (Zhang et al., 2025b) or embodied systems (Mendez-Mendez et al., 2023). The space of environments, tasks, and facts an agent encounters at inference far exceeds anything a finite training pipeline can prepare for. Following the "Big World" perspective (Sutton et al., 2023), the world is much more complex than an agent can model, making strong priors (i.e. knowledge from pretraining) a gross approximation of reality. Agents must continue to learn at runtime from a stream of observations, especially under practical compute and memory budgets. As a result, deep learning models must move beyond offline training.

We argue that transformers provide a practical starting point to solving lifelong learning (Biesialska et al., 2020). With attention, transformers exhibit a weak form of online adaptation through *in-context learning*. Conditioned on a short prompt, transformers adjust their behavior based on examples and instructions at inference time. However, softmax attention currently prohibits transformers from processing arbitrarily long contexts, preventing true long-horizon thinking.

To support lifelong in-context learning, we advocate for rethinking attention as an online learner. Building upon the lens of test-time regression (Wang et al., 2025), attention learns the relationship between keys and their associated values from the context at test-time. The set of key-value pairs forms an online training set, with the query as an unlabeled test point. In our generalization, softmax attention estimates the underlying key-to-value map with a nonparametric regressor (Nadaraya-Watson kernel estimation (Nadaraya, 1964; Watson, 1964)). This constrains softmax attention to shorter horizons. It stores past examples in a key-value cache, causing inference costs to grow with context length. Sparse alternatives (Nawrot et al., 2025) may bound the size of the KV cache. However, they evict tokens entirely rather than merge past memories. Sparse methods can catastrophically fail at simple, long-horizon tasks like variable tracking.

Our position is that extending in-context learning in transformers to lifelong settings requires *parametric forms of attention*. These mechanisms perform parametric regres-

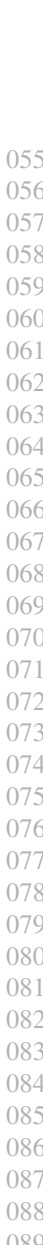

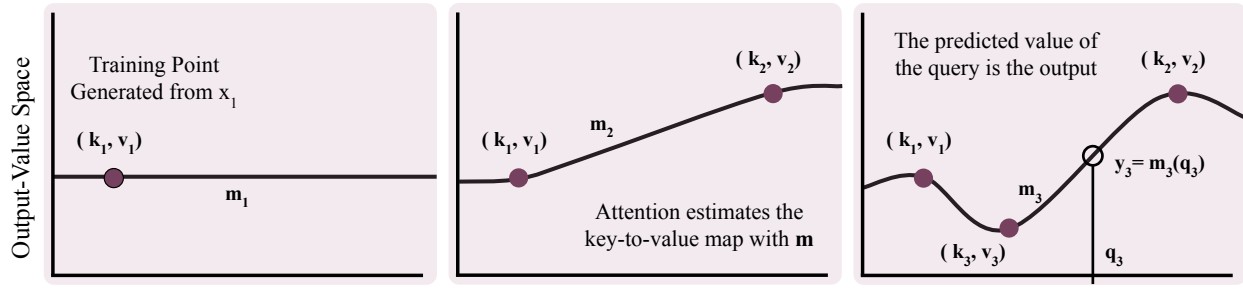

*Figure 1.* Attention as Test-Time Regression (Wang et al., 2025). Across three time steps, we illustrate how attention generates self-supervised training pairs and sequentially fits an estimator $m_t$. The output of attention at any given time is the prediction of the query.

sion over the past keys and values, effectively replacing the KV cache with a fixed-sized representation of past context. This structure appears across recent work on linear attention (Katharopoulos et al., 2020; Schlag et al., 2021; Yang et al., 2024b), state-space models (Dao & Gu, 2024), and fast-weight memories (Beck et al., 2024). In particular, methods that perform gradient descent online to optimize the estimate (denoted test-time training (Sun et al., 2024)) are the most promising candidates for lifelong learning (Behrouz et al., 2024; 2025a; Zhang et al., 2025a; Behrouz et al., 2025c). Despite progress, these approaches still face open problems in update efficiency (Zhang et al., 2025a), memory capacity (McDermott et al., 2025), and objective design (von Oswald et al., 2025; Behrouz et al., 2025b). Rather than proposing yet another mechanism, we organize these gaps into under-addressed questions intended to steer test-time training toward long-horizon agents.

Parametric forms of attention are a natural evolution in a rich history of adaptive filtering (Widrow, 1971), kernel regression (Nadaraya, 1964; Watson, 1964), and associative memory (e.g., Hopfield networks) (Hopfield, 1982), through fast-weight programming (Schmidhuber, 1992) and recurrent sequence models (Hochreiter & Schmidhuber, 1997). Transformers' query-key-value representation created a practical and parallelizable platform to bring old ideas to modern hardware. With this position paper, we aim to draw in researchers from self-supervised learning, reinforcement learning, and continual learning to help identify the right objectives, update rules, and architectures for long-horizon in-context learning.

## 2. Attention as Test-Time Regression

As a preliminary, we briefly explain traditional views of attention and formally introduce our perspective on attention as an online-learning algorithm. Next, we categorize past methods into nonparametric and parametric forms of attention.

### 2.1. Defining Attention

Transformers process a sequence of input tokens $\{x_t\}_{t=1}^{n}$, where $x_t \in \mathbb{R}^d$ (Vaswani et al., 2017). For each attention head, the input tokens are transformed into three representations—queries, keys, and values—via learned linear projections $\mathbf{W}_q, \mathbf{W}_k \in \mathbb{R}^{d \times d_k}$ and $\mathbf{W}_v \in \mathbb{R}^{d \times d_v}$ with

$$\underbrace{q_t = \mathbf{W}_q \, x_t}_{\text{query}}, \quad \underbrace{k_t = \mathbf{W}_k \, x_t}_{\text{key}}, \quad \underbrace{v_t = \mathbf{W}_v \, x_t}_{\text{value}}. \quad (1)$$

In causal softmax attention, the output at time $t$ is computed as

$$y_t = \frac{\sum_{j=1}^{t} \exp(q_t^\top k_j / \sqrt{d_k}) v_j}{\sum_{j=1}^{t} \exp(q_t^\top k_j / \sqrt{d_k})} \in \mathbb{R}^{d_v}. \quad (2)$$

This is typically interpreted as a local average of past values, weighted by how similar their associated key is to the query.

---

**Generalization of Attention**

We view attention as an online learning algorithm that recurrently estimates an unknown, non-stationary function, $m_t^* : \mathbb{R}^{d_k} \to \mathbb{R}^{d_v}$. The underlying function exactly maps keys to their associated values within a given context: $v_t = m_t^*(k_t)$. At inference time, attention receives a stream of self-supervised training examples $\{(k_j, v_j)\}_{j \leq t}$ generated by the transformer. Attention estimates the key-to-value relationship, then predicts the value associated with the query $q_t$ as the output $y_t$. This decomposes inference into an online learning problem:

Online training set: $(k_1 \to v_1), (k_2 \to v_2), .., (k_t \to v_t)$

Unlabeled test point: $(q_t \to ?)$

Formally, attention constructs an estimator $m_t : \mathbb{R}^{d_k} \to \mathbb{R}^{d_v}$, such that

$$m_t(k_t \mid \{(k_j, v_j)\}_{j \leq t}) \approx m_t^*(k_t) = v_t, \quad (3)$$

with hopes to generalize to unseen queries,

$$y_t = m_t(q_t \mid \{(k_j, v_j)\}_{j \leq t}) \approx m_t^*(q_t). \quad (4)$$

---

We visualize this interpretation in Figure 1. Under this view, attention is a form of test-time regression over keys and values (Wang et al., 2025).

We intentionally separate attention as a special kind of recurrent neural network (RNN)[1] that 1) explicitly forms self-supervised labels (key-value pair), 2) learns to predict one view from another (key-to-value estimation), and 3) predicts the association of a query $\boldsymbol{q}_t$ (the output $\boldsymbol{y}_t$). This structure continually generates and solves its own subtasks for the transformer to achieve its overarching goals (e.g. next-token prediction) (Sutton et al., 2023). The KV cache forms a hidden state, with updates driven by new observations to improve the estimation of $m_t^*$.

In a related perspective, $m_t$ acts as an associative memory system (Hinton & Anderson, 1989; Zhong et al., 2025) with each key-value pair serving as an associative memory. Building on ideas similar to Hopfield networks (Hopfield, 1982), this perspective (Iatropoulos et al., 2024) has inspired a resurgence of memory architectures across modern attention mechanisms (Behrouz et al., 2025c).

### 2.2. Nonparametric vs. Parametric

We classify attention mechanisms by how they represent and update the key-to-value estimators. Nonparametric forms of attention use nonparametric methods (e.g. nearest neighbors), leveraging infinite degrees of freedom and often unbounded inference costs. These approaches make minimal assumptions on $m_t^*$ as they do not construct a set of parameters, often using the training data as "parameters". As the

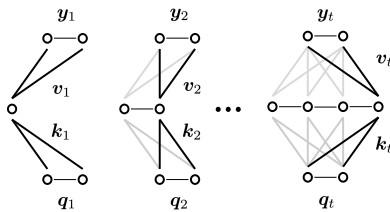

*Figure 2.* Classified as nonparametric, softmax attention uses the "training data" (key-value pairs) to form the estimator, leading to unbounded growth. Softmax attention can be viewed as an MLP with weights constructed from keys and values.

most popular instance of nonparametric attention, softmax attention corresponds to using Nadaraya-Watson kernel regression with an RBF kernel when queries and keys have unit 2-norm.

**Theorem 2.1** (Wang et al. 2025; Sun et al. 2024; Zhang et al. 2023; Han et al. 2023). *For a query $\boldsymbol{q}_t \in \mathbb{R}^{d_k}$ and context*

---

[1] We define RNNs as a general sequentially-updating function $f$ that maps an input and hidden state $(\boldsymbol{x}_t, \boldsymbol{h}_t)$ to an output with an updated hidden state $(\boldsymbol{y}_t, \boldsymbol{h}_{t+1})$. The hidden state is not restricted to any form, nor are other representations of the input enforced.

$\{(\boldsymbol{k}_j, \boldsymbol{v}_j)\}_{j \leq t}$ *with* $\|\boldsymbol{q}_t\|_2 = \|\boldsymbol{k}_j\|_2 = 1$, *softmax attention is equivalent to a Nadaraya–Watson estimator*

$$m_t(\boldsymbol{q}_t) = \frac{\sum_{j=1}^t \mathcal{K}_h(\boldsymbol{q}_t - \boldsymbol{k}_j)\,\boldsymbol{v}_j}{\sum_{j=1}^t \mathcal{K}_h(\boldsymbol{q}_t - \boldsymbol{k}_j)}, \tag{5}$$

*with kernel* $\mathcal{K}_h(\boldsymbol{q} - \boldsymbol{k}) = \exp(\boldsymbol{q}^\top \boldsymbol{k}/\sqrt{d_k})$ *and bandwidth* $h = 2\sqrt{d_k}$.

This regressor computes $m_t(\boldsymbol{q}_t)$ as the average of all values in a local neighborhood around $\boldsymbol{q}_t$, with the neighborhood shaped by the kernel $\mathcal{K}_h(\boldsymbol{q}_t - \boldsymbol{k}_j)$. Other nonparametric methods also exist, such as local polynomial regression (Henderson, 1916) or kernel ridge regression, though they are typically more computationally expensive. As another example, nonparametric attention could be formulated through kernel ridge regression by defining

$$\boldsymbol{y}_t = \exp\left(\frac{\boldsymbol{q}_t \mathbf{K}_t^\top}{\sqrt{d_k}}\right) \left(\exp\left(\frac{\mathbf{K}_t \mathbf{K}_t^\top}{\sqrt{d_k}}\right) + \lambda \mathbf{I}\right)^{-1} \mathbf{V}_t, \tag{6}$$

where $\mathbf{K}_t = [\boldsymbol{k}_1, \ldots, \boldsymbol{k}_t] \in \mathbb{R}^{t \times d_k}$, $\mathbf{V}_t = [\boldsymbol{v}_1, \ldots, \boldsymbol{v}_t] \in \mathbb{R}^{t \times d_v}$, the regularization coefficient $\lambda > 0$, and the exponential is computed elementwise.

Parametric forms of attention bound the degrees of freedom associated with the estimator $m_t$ by using parametric regression. These estimate the key-to-value map with a finite number of online parameters, $\theta_t \in \mathbb{R}^p$ for some constant $p$, with

$$m_t(\,\cdot\,|\theta_t) := m_t(\,\cdot\,|\{(\boldsymbol{k}_j, \boldsymbol{v}_j)\}_{j \leq t}) \tag{7}$$

Parametric attention encompasses many past approaches such as linear attention, state-space models, fast-weight programmers, and test-time training layers. In these cases, $\theta_t$ can be defined as the vectorized hidden states with $p$ being the maximum total number of elements in the states. Our definition intentionally includes sparse mechanisms that

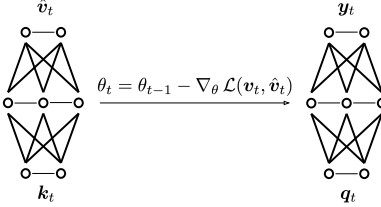

*Figure 3.* As an example of parametric attention, we visualize $m_t$ as a fixed-sized MLP that learns the key-to-value associations via test-time training.

only store a finite amount of tokens. For example, sliding window attention (Child, 2019) with window size $c$ has a bounded set of parameters with

$$\theta_t = \text{vectorize}(\{(\boldsymbol{k}_j, \boldsymbol{v}_j)\}_{j=t-c+1}^t). \tag{8}$$

Sparse selection (sliding window) and explicit gradient descent (test-time training) are only two of numerous ways to optimize $\theta_t$. Alternatives such as Bayesian Optimization (Kushner, 1964) could be explored.

## 3. Limitations of Nonparametric & Sparse Attention

Under fixed-hardware constraints, nonparametric estimators cannot be used as their effective capacity grows with the number of observed samples. We observe this clearly with softmax attention, as each new key–value pair $(\boldsymbol{k}_t, \boldsymbol{v}_t)$ is appended to the KV cache in order to estimate the map $m_t^*$. As inference costs grow beyond what hardware can support, past observations must be discarded or approximated.

While sparse attention methods (both the parametric and nonparametric kind) are an attractive way to extend the context window for modern LLM applications, they are not solutions to lifelong learning. These approaches retain only a subset of key-value pairs, evicting "unimportant" tokens from the cache. These methods fundamentally fail on any task that requires modeling of the *whole* context. Consider a sequence of updates

$$x \leftarrow 5, \quad y \leftarrow x + 5, \quad x \leftarrow 3, \quad z \leftarrow y + x.$$

Answering a query about $z$ requires incorporating every prior update. Dropping any assignment (e.g. $x \leftarrow 3$) renders the query unanswerable, regardless of how accurately the remaining tokens are recalled. Instead, the information from past tokens should be abstracted and compressed into a finite-dimensional representation.

As a result, attention mechanisms that support lifelong in-context learning must have:

- **Expressive in-context memory.** The online function, $m_t$, must model the whole context (eliminating any token-sparse methods). $m_t$ must recall specific events (retrieve values from past keys $m_t(\boldsymbol{k}_j) \approx \boldsymbol{v}_j$) and understand general concepts (generalize to queries with similar semantics as keys).

- **Per-token inference costs independent of context length.** The lifetime of these models is not known in advance. In-context learners must at least be able to process arbitrarily long contexts, eliminating nonparametric estimators for $m_t$.

- **Parallelizable updates.** For widespread adoption, methods must efficiently utilize modern hardware (Hooker, 2021) such as GPUs/TPUs, eliminating traditional RNNs. We expect these to at least incorporate chunkwise-parallelism (Yang et al., 2024c).

## 4. Test-Time Parametric Regression as a Solution

As the default solution, eviction-less parametric attention methods remain the only candidates for lifelong in-context learning. In this section, we argue that *test-time training* methods are the most promising lifelong learners. We formally define these methods, discussing different design patterns and limitations.

### 4.1. Online Objectives

Parametric forms of attention with test-time training define a differentiable loss function, $\mathcal{L}$, to model all contextual information with $\theta_t$. These methods can form higher-level abstractions that generalize as the context length increases.

To encourage $m(\boldsymbol{k}_t|\theta_t) \approx \boldsymbol{v}_j$, the online parameters $\theta_t$ are updated via gradient descent,

$$\theta_t = \theta_{t-1} - \nabla_\theta \mathcal{L}(\theta_{t-1}, \boldsymbol{k}_t, \boldsymbol{v}_t), \tag{9}$$

or similar optimization step. Choices for $\mathcal{L}$ are commonly the Hebbian rule (Hebb, 2005)

$$\mathcal{L}(\theta_{t-1}, \boldsymbol{k}_t, \boldsymbol{v}_t) = -\langle m(\boldsymbol{k}_t \,|\, \theta_{t-1}), \boldsymbol{v}_t \rangle, \tag{10}$$

Delta rule (Widrow & Hoff, 1988; Prados & Kak, 1989; Schlag et al., 2021)

$$\mathcal{L}(\theta_{t-1}, \boldsymbol{k}_t, \boldsymbol{v}_t) = \|m_t(\boldsymbol{k}_t|\theta_{t-1}) - \boldsymbol{v}_t\|_2, \tag{11}$$

or Omega Rule (Behrouz et al., 2025a)

$$\mathcal{L}(\theta_{t-1}, \{\boldsymbol{k}_j, \boldsymbol{v}_j\}_{t-c}^t) = \sum_{j=t-c}^{t} \gamma_j \|m_t(\boldsymbol{k}_j|\theta_{t-1}) - \boldsymbol{v}_j\|_2. \tag{12}$$

Though, new update steps are an active area of research. Input-dependent learning rates (Schlag et al., 2021), weight decay (gating) (Sun et al., 2023; Yang et al., 2024b; Dao & Gu, 2024; Yang et al., 2024a), momentum (vanilla (Behrouz et al., 2024) or orthogonalized (Jordan et al., 2024; Behrouz et al., 2025a; Zhang et al., 2025a)), and higher-rank gradients (Siems et al., 2025; Behrouz et al., 2025a) have also been incorporated into these update rules.

### 4.2. Parametric Functions

The family of parametric functions for $m_t$ is also under exploration. Most literature represents $m_t$ as a linear function. Notably, linear attention (Katharopoulos et al., 2020), uses the Hebbian rule with

$$m_t(\boldsymbol{q}_t|\theta_t) = \theta_t \boldsymbol{q}_t, \quad \theta_t = \theta_{t-1} + \boldsymbol{v}_t \boldsymbol{k}_t^\top \in \mathbb{R}^{d_v \times d_k}. \tag{13}$$

The recent revival of state-space models (Gu et al., 2021; Gu & Dao, 2024; Dao & Gu, 2024) also leverages linear functions.

On one hand, linear maps provide flexibility when parallelizing the online parameter updates over time (Yang et al., 2024c), though they fall short in memory capacity (McDermott et al., 2025). The number of orthogonal key-value pairs that can be stored is bounded by the rank of $\theta_t$. This can lead to degraded forms of long term memory, crucial for lifelong learning. On the other hand, online MLPs can learn much more complex key-to-value relationships with a relatively small footprint (Sun et al., 2024; Behrouz et al., 2024; Zhang et al., 2025a; Behrouz et al., 2025a), though these are expensive to update. Updating $\theta_t$ may require multiple backpropagation steps to fit the incoming KV pair.

Hybrid mechanisms can combine these approaches to fully leverage chunkwise-parallelism. In practice, fully parallelizing the context is not required under fixed hardware budgets. Only a subsequence large enough to saturate available VRAM needs to be parallelized. This view naturally motivates hybrid short- and long-term memory systems. For example, LaCT (Zhang et al., 2025a) pairs a fast local mechanism (sliding window attention) for recent context with a slow-updating MLP memory for long-term storage, updating the long-term memory once per chunk.

Moving forward, deep nonlinear parametric memories may present the strongest form of in-context learning. With test-time training, these can compress vast experience into a fixed-size latent state. Over long horizons, the estimator continually improves its high-level representation of the context, serving as an expressive in-context memory system. With the help of hybrid attention mechanisms, these can be trained fast with chunkwise-parallelism and constant per-token inference costs.

# 5. Open Questions and Research Directions

While the class of parametric attention mechanisms contains the solution to lifelong in-context learning, current methods still fall short in efficiency, capacity, and objective design. We highlight a small set of open questions that we believe will determine whether test-time learning mechanisms become practical for long-horizon agents.

**Question #1**

> *If the goal is not to memorize key-value pairs, what is the greater online objective? How should objectives incorporate regularization?*

As with most machine learning, the training loss is only a proxy; the ultimate goal is to perform well on the test set. To translate this to attention and test-time training, mapping past keys to their values is not the end goal; $m_t$ must map the query to the correct output $\boldsymbol{y}_t$. The "correctness" of the output is ultimately determined by the transformer and how well it solves the offline objective (next-token prediction). While this distinction seems obvious, this emphasis on generalization opens up novel insights on how the online objective should be designed.

In this section, we argue that forms of regularization in the online objective are needed. Gating mechanisms, which can be viewed as online weight decay, are the most notable forms of regularization as they forget old information to make space for new knowledge. Additionally, we observe implicit forms of regularization in both softmax attention and multi-headed linear attention, making them better online learners. We question how future algorithms can incorporate explicit forms of regularization in new memory architectures.

## 5.1. Case Study: Softmax Attention

Softmax Attention does not achieve perfect in-context recall. If $m_t(\boldsymbol{k}_t) \approx \boldsymbol{v}_t$ was the ultimate objective, then this problem is trivial: *just look up $\boldsymbol{k}_t$ in the KV cache and retrieve its value.* Mathematically, tightening the bandwidth of softmax attention's RBF kernel recovers this nearest-neighbors retrieval,

$$\lim_{h \to 0} m_t(\boldsymbol{k}_i) = \lim_{h \to 0} \frac{\sum_{j=1}^{t} \mathcal{K}_h(\boldsymbol{q}_t - \boldsymbol{k}_j)\boldsymbol{v}_j}{\sum_{j=1}^{t} \mathcal{K}_h(\boldsymbol{q}_t - \boldsymbol{k}_j)} = \boldsymbol{v}_i \quad (14)$$

for $\mathcal{K}_h(\boldsymbol{q}_t - \boldsymbol{k}_j) = \exp(\boldsymbol{q}_t^\top \boldsymbol{k}_j(\frac{2}{h}))$. It's simple to see that this leads to catastrophic overfitting (Barzilai et al., 2025), since the estimator does not generalize beyond the past values. The query is mapped to the value of its nearest neighbor,

$$\lim_{h \to 0} m_t(\boldsymbol{q}_t) \in \{\boldsymbol{v}_j\}_{j \leq t}. \quad (15)$$

Softmax attention (as opposed to "hardmax") provides a more generalizable estimate the key-to-value map, with respect to the query, though it does not perfectly interpolate the "online training data".

## 5.2. Case Study: Multi-Headed Linear Attention

It is common knowledge in the field of linear attention and SSMs use higher-dimensional hidden states for better in-context abilities. However, this is not exactly a rigorous rule-of-thumb. There is more nuance than the number of online parameters. For evidence, we look towards the role of multi-headed attention in linear attention.

Increasing the number of attention heads in a linear attention layer decreases the number of online parameters. An $h$-headed, $d$-dimensional linear attention layer estimates a different linear map for each head. For each head, $i$, and its linear map, $\theta_t^{(i)} \in \mathbb{R}^{d_k \times d_v}$, the output is defined as

$$\boldsymbol{y}_t^{(i)} = m^{(i)}(\boldsymbol{q}_t^{(i)}|\theta_t^{(i)}) = \theta_t^{(i)} \boldsymbol{q}_t^{(i)} \in \mathbb{R}^{d_v} \quad (16)$$

such that $d = d_k h = d_v h$. The heads' outputs are eventually concatenated and passed through a projection matrix $\mathbf{W}_{\text{proj}} \in \mathbb{R}^{d \times d}$. As a result, multi-headed linear attention constructs the overall key-to-value estimate as a block diagonal matrix:

$$\boldsymbol{y}_t = \left[ \boldsymbol{y}_t^{(1)}, \boldsymbol{y}_t^{(2)}, \ldots, \boldsymbol{y}_t^{(h)} \right] \tag{17}$$

$$= \left[ \theta_t^{(1)} \boldsymbol{q}_t^{(1)}, \theta_t^{(2)} \boldsymbol{q}_t^{(2)}, \ldots, \theta_t^{(h)} \boldsymbol{q}_t^{(h)} \right] \tag{18}$$

$$= \begin{bmatrix} \theta_t^{(1)} & 0 & \ldots & 0 \\ 0 & \theta_t^{(2)} & \ldots & 0 \\ \vdots & \vdots & \ddots & \vdots \\ 0 & 0 & \ldots & \theta_t^{(h)} \end{bmatrix} \begin{bmatrix} \boldsymbol{q}_t^{(1)} \\ \boldsymbol{q}_t^{(2)} \\ \ldots \\ \boldsymbol{q}_t^{(h)} \end{bmatrix} \tag{19}$$

$$= \theta_t \boldsymbol{q}_t. \tag{20}$$

The number of active parameters in $\theta_t$ is $d_k d_v h = d^2/h$. Additional heads regularize $\theta_t$, preventing the estimator from fully fitting to the context.

In the single-headed case (with $d_k = d$), we observe that $m_t$ has a simple closed-form solution for estimating $m_t^*$. If the key generating function, $\boldsymbol{k} = \mathbf{W}_k \boldsymbol{x}$, is invertible, then

$$\exists \mathbf{W}_k^{-1} \implies \boldsymbol{x}_t = \mathbf{W}_k^{-1} \boldsymbol{k}_t \tag{21}$$

$$\implies (\mathbf{W}_v \mathbf{W}_k^{-1}) \boldsymbol{k} = \boldsymbol{v}, \quad \forall (\boldsymbol{k}, \boldsymbol{v}) \tag{22}$$

$$\implies m_t^*(\boldsymbol{q}) = (\mathbf{W}_v \mathbf{W}_k^{-1}) \boldsymbol{q}. \tag{23}$$

The underlying key-to-value map is a linear function with $\theta_t^* = (\mathbf{W}_v \mathbf{W}_k^{-1})$. Crucially, the solution no longer depends on the context $(\boldsymbol{x}_1, \boldsymbol{x}_2, \ldots)$. Therefore, the perfect estimator (w.r.t the online objective) is static and does not incorporate any information from the prompt. This case study shows how converging to lower-loss solutions (i.e. using "better" optimizers) does not make a better attention mechanism. Instead, more difficult tasks or forms of regularization are needed to ensure $m_t$ is *useful*.

To measure the generalization behaviors of $m_t$ in practice, we suggest tracking the online loss for future key-value pairs. If $m_t$ can model future associations, $(\boldsymbol{k}_{t+i}, \boldsymbol{v}_{t+i})$ for $i > 0$, then the estimator may understand the underlying sequence that our context is a part of. Of course, this still requires the online task to be sufficiently difficult.

In summary, the goal of the estimator is not only to memorize past relationships, $m_t(\boldsymbol{k}_j) \approx \boldsymbol{v}_j, \forall j \leq t$, but to also generalize to unseen queries, $m_t(\boldsymbol{q}_t)$. Associative memory systems that fit to the online training set can recall exact historical information, which is the foundation for episodic memory. However, lifelong in-context learning also requires understanding semantic information (general knowledge of the world), which can be viewed as abstractions of past events. To recall a fact, such as knowing the earth revolves around the sun, efficient memory systems should not have to trace back to the time they first learned this fact. Attention mechanisms should also broadly store knowledge after learning it in-context.

## Question #2

> *How can online loss functions efficiently contain longer-horizon signals?*

Besides fitting to new observations and retaining earlier associations, lifelong learning requires identifying trends that emerge over time. For the next generation of parametric methods, long-horizon information needs to be incorporated in the online parameters. This motivates our question about how to best model the whole context under a fixed compute budget.

Early parametric methods, such as linear attention, Deltanet, and Retnet (Sun et al., 2023), only use the current key-value pair to update the online parameters. Methods with instantaneous loss functions enforce $m_t(\boldsymbol{k}_t) \approx \boldsymbol{v}_t$, but these can only *hope* that $m_t$ retains past knowledge ( $m_t(\boldsymbol{k}_j) \approx \boldsymbol{v}_j$ for $j < t$). Especially with linear methods, McDermott et al. observe that instantaneous updates can unintentionally overwrite past information, leading to catastrophic forgetting. While instantaneous objectives are efficient, these are likely suboptimal for memory retention.

We observe two common trends in recent literature that inject longer-horizon information: batched updates and auxiliary states. While we believe these ideas are on the right track, we note that current parametric attention methods still use short-horizon updates. This remains an open problem on that path to true lifelong in-context learning.

**Batched Updates.** Memory updates can be performed over a batch of key-value pairs. Behrouz et al. use a sliding window of the last $c$ tokens, optimizing a weighted regression loss with decay terms $\gamma_i$,

$$\sum_{i=t-c+1}^{t} \gamma_i \left\| m(\boldsymbol{k}_i | \theta_{t-1}) - \boldsymbol{v}_i \right\|_2^2, \tag{24}$$

denoted the Omega rule. Moving beyond a sliding window, other sparse attention criteria can select which tokens should be cached and used again for future updates (akin to memory replay in continual learning (Rolnick et al., 2019)). For example, key-value pairs in the current batch that are not properly estimated by $m_t$ can be cached for the next batch (McDermott et al., 2025).

**Auxilliary States.** A second approach is to leverage additional memory states for a more informative update. As an example, MesaNet (von Oswald et al., 2025) implements a

second-order optimizer over the cumulative loss,

$$\sum_{j=1}^{t} \|m(\boldsymbol{k}_j|\theta_{t-1}) - \boldsymbol{v}_j\|_2^2 + \frac{1}{2}\mathrm{Tr}(\theta_{t-1}\,\Lambda\,\theta_{t-1}), \quad (25)$$

with $\Lambda$ acting as a quadratic regularizer. This loss is optimized by

$$\theta_t^{(1)} = \gamma_t \theta_{t-1}^{(1)} + \beta_t \boldsymbol{v}_t \boldsymbol{k}_t^\top \in \mathbb{R}^{d_k \times d_v} \quad (26)$$

$$\theta_t^{(2)} = \gamma_t \theta_{t-1}^{(2)} + \beta_t \boldsymbol{k}_t \boldsymbol{k}_t^\top \in \mathbb{R}^{d_k \times d_k} \quad (27)$$

$$\boldsymbol{y}_t = \theta_t^{(1)}(\theta_t^{(2)} + \Lambda)^{-1}\boldsymbol{q_t}. \quad (28)$$

Here, $\theta_t^{(1)}$ forms the base key-to-value map, with a gated linear attention learning rule (Yang et al., 2024b), and $\theta_t^{(2)}$ decorrelates past keys. The additional state improves key-to-value learning while still maintaining the efficiency of linear maps. This concept of accumulating historical information in alternative states is also present in momentum-based optimizers (Behrouz et al., 2024) (though on a much shorter horizon).

Combining both approaches, LaCT (Zhang et al., 2025a) and Atlas (Behrouz et al., 2025a) compute the online loss over the past chunk of tokens and use an additional momentum state with the Muon optimizer (Jordan et al., 2024). A hidden state $\theta_t^{(\text{state})}$ is updated with a momentum state $\theta_t^{(\text{mntm})}$, not the direct gradient, as in

$$\theta_t^{(\text{mntm})} = \gamma_t \theta_{t-1}^{(\text{mntm})} + \nabla_\theta \mathcal{L}(\theta, \boldsymbol{k}, \boldsymbol{v}) \quad (29)$$

$$\theta_t^{(\text{state})} = \theta_{t-1}^{(\text{state})} - \beta_t\,\mathrm{Orthogonalize}(\theta_t^{(\text{mntm})}) \quad (30)$$

$$\boldsymbol{y}_t = f(\boldsymbol{q_t}, \theta_t^{(\text{state})}). \quad (31)$$

Here, $f$ is some function parametrized by $\theta_t^{(\text{state})}$, and $\gamma_t$ still represents a decay term. The loss $\mathcal{L}$ is computed over the past sliding window / chunk of KV-pairs, and the momentum term $\theta_t^{(\text{mntm})}$ is orthogonalized with Newton-Schulz matrix iteration (Jordan et al., 2024; Bernstein & Newhouse, 2024; Björck & Bowie, 1971). Orthogonalization prevents any singular vector in the update from overwriting stored values in online parameters, similar to how low-rank updates observe less catastrophic forgetting (Biderman et al., 2024).

In summary, both batched updates and auxiliary states shown to improve contextual modeling (Behrouz et al., 2025a); though, there is still work to be done. We suggest researchers to draw inspiration from continual learning over data-streams. In particular, future attention mechanisms should make use of memory replay.

**Question #3**

> *Are linear self-supervised labels sufficient? How should keys and values be generated?*

In parametric attention, the online learner $m_t$ is trained on self-supervised pairs $(\boldsymbol{k}_t, \boldsymbol{v}_t)$ produced by the offline transformer. During pretraining, the transformer must learn to create informative online tasks that attention solves during inference. We question how the key-value observations should be generated, as this directly relates to what kind of information the online estimator learns. As a reminder, to scalably learn at test-time, the online tasks or subproblems cannot be hand-crafted and should minimize the amount of human priors (Sutton et al., 2023).

Most transformer implementations generate $\boldsymbol{k}_t$ and $\boldsymbol{v}_t$ as linear projections of $\boldsymbol{x}_t$, which makes the key-to-value map structurally simple. Nonlinear constructions can make the underlying map $m_t^*$ more complex and may force $m_t$ to capture higher-level structure from context. This can be performed through nonlinear feature maps such as $\boldsymbol{k}_t = \mathbf{W}_k^{(2)}\sigma\left(\mathbf{W}_k^{(1)}\boldsymbol{x}_t\right)$. Temporal information can be encoded into the key-value pair with positional embeddings (e.g. RoPE (Su et al., 2024)), changing the online estimation. Lastly, key-value pairs do not have to be functions of $\boldsymbol{x}_t$ alone. Short convolutions and other local mixing operations have commonly been used in recurrent sequence models (Allen-Zhu, 2025). This creates short-horizon observations for $m_t$ to estimate.

The ratio between self-supervised label (KV pairs) and test points (queries) is an underexplored design axis. Standard multi-headed attention produces one query and one key-value pair per head per token. Grouped-query attention (Ainslie et al., 2023) reduces the number of distinct key-value memories while keeping many queries, allowing multiple "questions" to be asked to same contextual model $m_t$. In the other direction, DeltaProduct updates $m_t$ with multiple key-value pairs from a token for a richer update (Siems et al., 2025). Notably, this enables state tracking behavior in linear models. Understanding this tradeoff likely depends on the parametric family used for $m_t$; however, this raises many fundamental questions about the role of multi-headed attention. For example, in deep, nonlinear parametric methods are multiple estimations of the context needed or can expressive forms of $m_t$ suffice with a single head?

The current literature lacks a fundamental understanding of what this self-supervised task should look like. What properties of the generated $(\boldsymbol{k}_t, \boldsymbol{v}_t)$ pairs make the resulting online learner useful for the outer objective? Is there a relationship between the difficulty of this task and its usefulness? We expect progress here to draw from self-supervised

learning, where the central question is how to construct informative views and prediction targets, such as work on latent prediction like JEPA (Assran et al., 2023).

## 6. Alternative Views

This position paper intends to shift the focus (a finite resource in the research community) towards parametric forms of attention. With this, we expect some controversy, especially around 1) the sufficiency of current transformers 2) the scalability of parametric attention methods and 3) the novelty of this position.

**(a)** *Modern transformers are a sufficient backbone for agents when augmented with retrieval, tool use, recursive prompting, etc.*

These augmentations build on top of foundation models by storing "agentic memories" which summarize past experience into notes (Xu et al., 2025; Zeng et al., 2024). While these scaffolding approaches have scaled LLMs from question-answer tasks to complex code generation, this cannot be extended to lifelong learning. The base model must be able to understand and learn from new, out-of-distribution observations. To perform complex tasks and understand a lifetime of knowledge, experiences need to be modeled in an internal, latent representation, not a notebook.

**(b)** *There's little evidence that these approaches even scale to hundreds of billions (or trillions) of parameters!*

We agree that these methods (SSMs, linear attention, test-time training layers) have been mostly evaluated at smaller scales (8B parameter LLMs), often under academic training budgets. However, we argue that this response should motivate further research in the area, not pre-emptively halt its progression. Following the spirit of our position, this should urge researchers to isolate the bottlenecks that prevent scaling, studying them directly, rather than dismiss these methods outright.

**(c)** *This position is not new; SSMs, linear attention, and test-time training are all quite popular!*

We agree that these directions have gained traction in recent years. However, their framing and impact are not yet universal; much of the sequence-modeling literature treats SSMs and linear attention as alternatives to transformers, rather than an extension of it. Meanwhile, many practitioners still treat transformers as a static backbone and focus on via sparse attention and wrappers. Our goal of this paper is to translate the ideas of test-time training back into the world of transformers. We demonstrate that softmax attention is an online learner, performing a similar self-supervised process.

## 7. Conclusion

In this paper, we argued that lifelong in-context learning under fixed compute is a fundamental problem for AI systems. We framed attention as an online learner to make explicit what must change as we move toward long-horizon agents. Softmax attention represents experience nonparametrically by retaining past key-value pairs, so inference cost grows with context length. We advocated for *parametric forms of attention*, which learn from self-supervised observations of the context while maintaining constant per-token inference costs. In particular, we emphasized test-time training as a direct way to optimize parametric memories online.

Despite promising progress, current parametric approaches still fall short in update efficiency, memory capacity, objective design, and long-horizon training. Rather than proposing a new mechanism, we distilled these issues into a small set of open questions intended to guide the field toward long-horizon agents. We hope this perspective helps translate ideas across communities and encourages more direct work on attention mechanisms in lifelong settings.

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
