# OpenReview forum: "Position: Lifelong In-Context Learning Requires Parametric Forms of Attention"
_ICML.cc/2026/Position_Paper_Track — Submitted to ICML 2026 Position Paper Track_

### Official Review · Reviewer_tsfa · 2026-03-08

**Significance:** 3
**Argument Clarity:** 3
**Rating:** 4
**Confidence:** 2

**Questions:**

1. The paper identifies test-time training methods as the most promising lifelong learners。 Then, how does this approach integrate with existing training paradigms? For instance, in the context of training large language models, should test-time training be incorporated into existing pipelines (e.g., SFT, RLHF) or treated as a distinct, standalone training stage?

**Alternative Views Section:**

Yes

**Compliance With Llm Reviewing Policy A Conservative:**

Affirmed.

**Discussion Potential:**

3

**Paper Summary:**

This position paper argues that **lifelong in-context learning** in AI agents requires **parametric attention mechanisms**, addressing the limitation of nonparametric softmax attention (quadratic cost, unbounded KV cache growth) for long-horizon tasks. As in the community there is a shift toward agents operating for months/years (e.g., autonomous research systems) that need to learn online from infinite streams of novel information under fixed hardware budgets. The core contribution is framing attention as test-time regression—where nonparametric methods (softmax) use kernel estimation (Nadaraya-Watson) and parametric counterparts (linear attention, SSMs, test-time training layers) replace KV caches with fixed-size online-learnable parameters—along with key open questions to guide progress.

**Position:**

Yes

**Position In Title:**

Yes

**Related Work:**

3

**Strengths And Weaknesses:**

### Strengths
1. The paper offers an inspiring and thought-provoking perspective: it frames current attention mechanisms as a form of test-time regression over keys and values, arguing that nonparametric and sparse attention are inherently ill-suited for lifelong learning. By identifying test-time training methods as the most promising candidates for lifelong learners, the paper also courageously questions the sufficiency of transformers—long treated as the default backbone—against the real-world demands of lifelong learning, which align with the current trajectory of LLM development.
2. The paper raises fundamental, impactful open questions and research directions that address critical gaps in the field, providing valuable guidance for future exploration into long-horizon in-context learning.

### Weakness
1. The paper’s positions would be more compelling with minimal empirical experiments to validate its core claims.
2. The case studies in Section 5 feel somewhat redundant given the preceding discussions, offering few new insights. It would be more effective to focus on exploring additional potential solutions in the main text and relegate detailed analytical depth to the appendix.

**Support:**

3

---

> ### Author Rebuttal · Authors · 2026-03-31
>
> Thank you for your review and positive feedback.
>
> > “The paper’s positions would be more compelling with minimal empirical experiments to validate its core claims.”
>
> In our revision, to support our claims of the unsustainable costs of nonparametric approaches, we will add additional plots on the per-token inference cost. This will measure both FLOPs & VRAM usage across sequence lengths from 2^10 to 2^32, along with various curves for token dimensionality and precision. Alongside this, we will show how compute trends cannot keep up with the vision for lifelong inference.
>
> >“The case studies in Section 5 feel somewhat redundant given the preceding discussions, offering few new insights. It would be more effective to focus on exploring additional potential solutions in the main text and relegate detailed analytical depth to the appendix.”
>
>
> We will incorporate this feedback into the camera-ready draft by exploring more potential solutions.
>
> >“The paper identifies test-time training methods as the most promising lifelong learner. Then, how does this approach integrate with existing training paradigms? For instance, in the context of training large language models, should test-time training be incorporated into existing pipelines (e.g., SFT, RLHF) or treated as a distinct, standalone training stage?”
>
>
> Existing pipelines such as pretraining, SFT, and RLHF can be integrated with test-time-training layers such as Titans, Atlas, or LaCT, since these methods replace the softmax attention mechanism within a transformer. Test-time training is executed during each forward pass, while offline training still backpropagates through the online-learning module to optimize the surrounding offline parameters (e.g., $W_q$, $W_k$, …) in the usual way. When trained with today’s objectives and context lengths, however, this will generally still produce a model optimized for relatively short-horizon language modeling.
>
> Though, for true lifelong learning over million- or trillion-token horizons, we believe today’s pipelines are likely insufficient, since they are primarily optimized for standard language modeling over short contexts. Under this regime, much of the required knowledge must be stored in persistent model weights because the relevant information is not available in context. As horizons grow much longer, deployment itself becomes the dominant source of novel information, suggesting that pretraining may need to play more of a meta-learning role. This should prepare the model with the right priors for in-context learning, allowing it to acquire new knowledge and perform new tasks. How to pretrain such systems effectively for long-horizon task acquisition remains an open question.
>
> Our position paper focuses on the first step: how should a lifelong agent perform inference? This naturally leads to two further open problems: how should such an agent be trained and how should it be evaluated?

---

> > ### Author Rebuttal · Reviewer_tsfa · 2026-04-03
> >
> > Thanks for your reply. The concerns are partially resolved, and it is better with more (minimal) experiments.

---

### Official Review · Reviewer_H3fq · 2026-03-13

**Significance:** 4
**Argument Clarity:** 2
**Rating:** 4
**Confidence:** 4

**Questions:**

* For specific long-context tasks or inputs, nonparametric attention may simply be superior to finite parametric attention mechanisms. Do the authors believe we should be focusing on purely parametric attention? Or do hybrid methods, such interleaved SSM / full attention models, offer a practical best-of-both-worlds approach?
* Are there any empirical or theoretical arguments that support your claim that RAG + agentic workflows cannot fundamentally scale to lifelong learning? What specifically makes this framework inferior to internal parametric memory?
* Given the success in massively parallelizing softmax attention on modern GPUs, are novel hardware architectures required to scale beyond the bottlenecks facing current parametric attention models? Or could algorithmic and software advances be sufficient to make parametric methods scalable on current hardware?
* The authors state that sparse attention is non-parametric due to the truncated context. However, with casual LM future tokens lossily encode past tokens already. Could the authors further clarify why these KV representations cannot already be considered a form of parametric memory?

**Alternative Views Section:**

Yes

**Compliance With Llm Reviewing Policy A Conservative:**

Affirmed.

**Discussion Potential:**

4

**Final Justification:**

The rebuttal addressed most of my questions and I increased my rating to suit. Some elements of the paper's position remains weakly supported in my opinion, so I did not increase my score further.

**Paper Summary:**

This paper argues that parametric forms of attention will be required to unlock life-long context learning. The authors reframe attention as test-time regression and argue that an infinite KV cache is required to support arbitrarily long sequences. The paper outlines current limitations of parametric approaches, open research questions, and alternative views.

**Position:**

Yes

**Position In Title:**

Yes

**Related Work:**

4

**Strengths And Weaknesses:**

## Strengths
* Nice presentation, the work is enhanced by the figures and emphasized paragraphs.
* Excellent formal introduction of parametric vs. non-parametric forms of attention.
* The outline of open research questions are well grounded in current literature and identifies specific challenges yet to be addressed in the field. The specific identification of current trends and suggestions to draw inspiration from existing research niches from the ML community (i.e., continual learning) offer concrete, actionable take-aways.
* Reasonably strong review of alternative views.
* Taken as a whole, the paper synthesizes results from multiple subfields and connects approaches such as state-space models with test-time training and associative memory.

## Weaknesses:
* Ultimately, the paper lacks direct evidence to back up the posiiton that lifelong in-context *requires* parametric attention. The authors provide a convincing argument that the research community should continue to study and develop methods of parametric attention; however, it does not follow that there are no other viable strategies that could achieve the ultimate aim of lifelong in-context learning.
* The authors note that attention mechanisms that support lifelong context “must model the whole context”. However, we know that any given finite state will be strictly limited from modelling sufficiently complex contexts due to information theoretic limits. Given a finite attention workspace, how can contexts with a Kolmogorov complexity that exceeds the capacity of the parametric attention to model be accurately modelled? While the position argued is a strong one, incorporating the theoretical limits of parametric forms of attention may help ground expectations.
* Alternative view 6.a) is dismissed as “experiences need to be modeled in an internal, latent representation, not a notebook”. Given the finite capacity of any parametric form of attention, it seems likely that a combination of these approaches are required. After all, the success of “notebooks” in augmenting human intelligence is indisputable. The authors could clarify that these two approaches need not be at odds. Better internal representations can be augmented with external persistent storage and retrieval that offers a best-of-both-worlds approach.
* The authors cite the Hardware Lottery as a primary reason why current approaches have become popularized and note, as an alternative view, that parametric attention has yet to be scaled to frontier model size. To alleviate this, they urge researchers to isolate and study bottlenecks in parametric attention methods. However, aren’t these bottlenecks already well understood with respect to relatively small amounts SRAM available for fast state updates and memory bandwidth bottlenecks?
* Accuracy is paramount. Thus far, parametric forms of attention have yet to exceed the capabilities of nonparametric attention. For certain classes of long-context tasks, such as variable tracking and common word extraction, it seems SSM and their variants still lag significantly behind current nonparametric approaches. It remains unclear whether this accuracy gap can be overcome with finite parametric attention.
* No empirical data is provided that could strengthen the arguments.

**Support:**

2

---

> ### Author Rebuttal · Authors · 2026-03-31
>
> Thank you for the thorough review and feedback. We will try to be as descriptive on limited space.
>
> **W1**
> We will revise our draft to more rigorously define our statements. We will also add a defining assumption that in-context learning is the process of the transformer learning from context with attention.
>
> While there may exist other viable strategies in general for lifelong learning, our position states that lifelong *in-context learning* with transformers requires a bounded representation of past context, when considering practical, fixed-hardware budgets. Nonparametric attention mechanisms are not computable at lifelong context lengths. Thus, parametric attention mechanisms remain as the default solution.
>
> **Q4 / Q2 / W3**
> Sliding window attention is an example of parametric, sparse attention because both inference and the state update have bounded representations. Sparse selection methods that use the top-k most similar keys to the current query require some sort of KV-offloading. These have infinite degrees of freedom, thus they are nonparametric. In the setting with streaming key-value pairs, such as generation, we want memory costs to be constant, not growing.
>
> This also includes memory costs of offloading, so RAG---which is conceptually similar to sparse retrieval---must have a finite capacity as well in lifelong settings. To effectively utilize this limited capacity, latent representations of past context are more expressive than natural language. While RAG is not a solution to lifelong in-context learning in itself, we agree external tools can improve capabilities. We will update our language in 6.a).
>
> Additionally, see Sutton's Era of Experience [3] for Q2.
>
> **W2**
> Yes, a finite state is limited in expressivity, so it cannot learn arbitrarily complex tasks in context. In the infinite compute regime, nonparametric methods are powerful because they increase the modeling complexity to accommodate the data. Given that a finite state is required for a fixed hardware set up, we cannot do that. It is an open question on how to maximize the expressivity of the bounded cost, parametric functions (we suggest MLPs over linear maps), while balancing the bias-variance tradeoff. We will update section 4.2 with this discussion to ground expectations.
>
> **W4**
> We agree that the hardware bottlenecks (e.g. SRAM, memory bandwidth) are well understood; however, we intend for researchers to explore how parametric attention methods specifically interact with this. For example, on lines 233-242, larger chunksizes leverage more parallelism, but this can affect the solution of the test-time optimizer as it uses a larger “batch size” per update.
>
> Additionally, parametric mechanisms significantly lag behind transformers in implementation maturity. Softmax attention is well-optimized across generations of GPUs (ex: FlashAttention4 is specifically designed with Nvidia’s B200 in mind). As a result, widespread adoption of a new “attention” mechanism—even if they are fundamentally superior—can be incredibly challenging if developed in isolation. This is especially important as our generalization of parametric attention encompasses a broad function class.
>
> **W5 & Q1**
> In this work, we have our eyes set on extreme-context length scenarios with lifelong learning, rather than the current evaluation suite of long-context LLMs. We agree that, on tasks such as variable tracking / CWE for sequence lengths in the thousands of tokens, SSMs and recent test-time training variants can still be lacking, as softmax attention has superior recall in these settings. However, we are interested in such settings that softmax attention cannot even perform inference on (i.e. billions of tokens in context or more) like learning in the wild over years or decades [3]. Nonparametric forms of attention are not computably in such settings. Hybrid methods that replace some layers still have this fatal flaw for extreme context length inference. Our perspective is not whether parametric attention is good enough---as this is the only practically computable solution---but rather "how do we improve this the best we can?"
>
>
> **W6**
> In our revision, to support these claims of the unsustainable costs of nonparametric approaches, we will add additional plots on the per-token inference cost. This will measure both FLOPs & VRAM usage across sequence lengths from 2^10 to 2^32. We conservatively chose the maximum sequence length to approximate all the sensory inputs of a human over 1 second as a single token. Alongside this, we will show how recent improvements in hardware cannot keep up with the vision for lifelong inference.
>
> **Q3**
> Novel hardware architectures are not required. These mechanisms have been, and continue to be, designed with GPUs in mind [Yang, 2024c], providing the necessary recipe for scale.
>
> [3] Silver, David, and Richard S. Sutton. "Welcome to the era of experience." Google AI 1 (2025): 11.

---

> > ### Author Rebuttal · Reviewer_H3fq · 2026-04-03
> >
> > I thank the authors for their detailed rebuttal. Under the specific constraints of lifelong in-context learning, the arguments put forth are reasonably compelling and have good discussion potential. I have elected to raise my score.
> >
> > However, I remain unconvinced that continuous, parametric test-time learning is the only viable path forward. If the goal "is a practical solution for continual learning in AI agents". A plausible and scalable alternative is simply a finite tape with explicit read, write, and delete operations, paired with periodic offline updates. Using nonparametric attention with a tape, we can decouple short-term recall from long-term abstraction.
> >
> > The ambitious goal of extreme context is interesting. But current failures at comparatively "small" long context tasks guides my intuition that a nonparametric working memory *combined* with a finite cache (parametric or otherwise) may be a more practical solution in the near- to medium-term.

---

### Official Review · Reviewer_qDWj · 2026-03-14

**Significance:** 3
**Argument Clarity:** 3
**Rating:** 2
**Confidence:** 2

**Questions:**

I do not have specific questions, but very open to arguments clarifying possible confusions or rebutting the above outlined counter-arguments.

On a higher conceptual level, I am not even convinced of the the very first argument. Suppose for a moment, that softmax attention would be feasible with a context much longer than the lifetime of the system. Would it be then the right mechanism for lifelong learning? Or rather it is just one mechanism that has features of lifelong learning but not really rationally designed to serve that purpose?

**Alternative Views Section:**

Yes

**Compliance With Llm Reviewing Policy A Conservative:**

Affirmed.

**Discussion Potential:**

3

**Final Justification:**

The main proposition -- to research parametric forms of attention for improving efficiency of transformers and addressing lifelong in-context learning --  is an interesting, actual, and fruitful for discussion, in my view.

After the rebuttal, my standing issues with the paper are as follows:

1. As the central tool for discussing parametric attention models the paper advocates the view "Sec. 2. Attention as Test-Time Regression" and "Sec 4. Test-Time Parametric Regression as a Solution", mainly backed up by the Nadaraya-Watson estimator (5) which takes the form of the common attention (for normalized k,q). I believe this reasoning is not rigorous and has subtle flaws. The NW estimator solves the regression problem of the kind $m(x) = \mathbb{E}[y \mid x]$ by using kernel density as a plug-in in the conditional expectation. This regression problem formulation assumes i.i.d. observations (y,x). It is not the case for key-value pairs in transformers because they are not independent over tokens. Of interest is "a form of time-series regression" $m_t(q_t \mid \{k_j, v_j\}_{j\leq t})$, which however the paper **does not define in a rigorous way**. For the latter problem, occuring on-line and in-context, we do not have a distribution of input sequences and targets, with which a statistically meaningful **regression goal** could be set up, we only have one on-line sequence. Furthermore, the key-value pairs are also dependent across layers. This makes any test-time regression concept rather questionable. The proposition that the outer transformer training is what sets up the regression target is not formalized.

2. The paper mixes up in the reasoning the simple regression problem behind NW estimator and the one that is not formalized, as discussed above, which is **misleading**!. The main proposal is to view learning a (parametric) mapping from keys to values as a regression problem of the first simple kind but **interpret** it as a regression of the second complex and undefined kind. By an example of a functional transformer architecture where keys and values are identified, I argued that this regression goal maybe trivially achievable without useful outcomes. This implies that constructing other (parametric) attention learning mechanisms in the proposed paradigm is not sound, for instance "to encourage $m(k_t | \theta_t) \approx v$" makes no sense.

In my view, the above problems invalidates / make obsolete discussion and proposals in "Sec 4.1 Online Objectives", "Question #1: If the goal is not to memorize key-value pairs,what is the greater online objective? How should objectives incorporatere gularization?", "Question #3: Are linear self-supervised labels sufﬁcient? How should keys and values be generated?". And I think they make up for a major reason for rejection. To rectify the issue it might be needed giving up the paradigm of "test-time regression", which would change the whole paper.

At the same time, authors have argued that the regression view was already used in the literature in some fruitful way. I have not red these papers and not familiar with in-context learning, which I indicate by a low confidence level (2). If there is no support amongst reviewers for the points I brought up, please feel free to accept -- it is a good presentation and great potential for the discussion and let's see if the authors could improve the reasoning.

**Paper Summary:**

The paper discusses the approaches to address lifelong in-context learning. The starting point of the discussion is that transformers, can learn a lot through generalizing over the context, and thus with an unlimited context, could in principle learn and later use new information online. The problem then becomes entangled with that one of handling long context of transformers. It is argued that self-attention is a form of online-trained regressor and that from that point of view, parametric regressors of a kind, their respective learning objectives and rules should be advanced.

**Position:**

Yes

**Position In Title:**

Yes

**Related Work:**

3

**Strengths And Weaknesses:**

The motivation is really strong. Life-long learning would unlock many benefits. Even addressing the inefficiency of the current technology (full re-training cycles, huge memory and computation costs needed for inference with large context), appears very important. The paper is well-structured in terms of explaining the vision of the problem, linking the concepts, showing examples from the literature, and articulating open questions.

My position, however, is that the reasoning is not sufficiently rigorous and conceptually clear.

The "Attention is a Test-Time Regression" view draws parallels between softmax attention and e.g. Kernel regression with RBF Kernel. While the math fits (Theorem 2.1), the settings are different:
- The regression seeks to approximate a function over a distribution of examples $x$, in expectation, where RBF kernel corresponds to the prior assumption of smoothness of this distribution and makes sense when the data samples are i.i.d.
- The attention is a non-linear filter of highly-interdependent past representations (k_i,v_i). The mathematical similarity with the regression does not imply that the function and learning problems are related. *It does not imply* that in order to implement useful conditional dependencies $m(y_t | (k_j, v_j)_j)$, other regression formulations of values from keys and the associated learning procedures can be conceptually meaningfully employed.

Based on the above (mis)conception, the paper mixes predicting values $v_t$ from keys $k_t$ (which may be trivial) and the problem of predicting $y_t$ from $q_t$, which is hard. This is why I think the presented generalization of attention is conceptually misleading. The examples present SSM as "key to value maps" but they are not as equation (7) RHS, dependent on $theta$, actually exposes. In the discussion of  "Question 1", the paper concludes that "mapping past keys to values is not the end goal, $m_t$ must map the query to the correct output $y_t$". Then the initial formalism was not correct in the first place?

The splitting between keys and values is a rather arbitrary design choice of the current technology. Without loss of generality transformers could merge keys and values into one vector (i.e. concatenate the two and let query vector be concatenated with a vector of zeros to multiply the value part). This underlines that the problem of predicting "values from keys" is not of much interest and cannot be a solid basis for life-long learning. The paper admits and discusses this issue in "Question 3", but proposes heuristic mitigations without a solid conception of the in-context learning. Similarly, I think, interpreting the representations kept by the sliding window attention as parameters of test-time learning is a stretch.

The paper is not very clear about what is lifelong learning, in-context learning and their goals. These notions seem to me to be mixed or even identified with the problem of handling large inference context. Their limitations are also not very clear as, for instance, SSMs, or recurrent networks or Performer like models all have virtually unlimited context (but admittedly hold only a limited representation of it).

To summarize, I see the case for parametric attention methods, in order to generalize for longer contexts, but it is rather known. But I do not see a strong and a clear conception for life-long learning via a kind of auxiliary test-time regression "keys->values" produced by the model. The questions posed by the paper such as need for regularization, are all secondary from that perspective.

**Support:**

2

---

> ### Author Rebuttal · Authors · 2026-03-31
>
> We will be updating our language in the draft to better explain our generalization of attention and add our definition of lifelong learning following [3]. First, we would like to provide a few clarifying statements to form a better foundation:
> - We define attention as a function $m_t$ of the current query and all past key/value pairs. This estimates the underlying relationship between keys and values.
> - The overall transformer solves for some objective during offline pretraining (or SFT/RLHF), such as next-token prediction. This outer loop objective optimizes $W_q$, $W_k$, etc., and the feedforward network parameters.
> - The transformer generates the set of key-value pairs (online training dataset) and creates the current query as an unlabeled, online test point. On every forward pass at every time step, attention solves an inner-loop objective by fitting to the “online training set”. Then, attention predicts the associated value of the query (denoted as the output $y$). The queries and keys live in the same space in $\mathbb{R}^{d_k}$. The values and output live in $\mathbb{R}^{d_v}$.
> - This inner-loop, online optimization occurs regardless if the transformer is performing pretraining or inference.
> - The outer-loop, **offline** optimization process teaches the transformer how to create a valuable, **online** dataset. If the transformer generates a trivial dataset (ex: make k=v) then the **online** estimator $m_t$ does not learn meaningful information from the context and would not provide a useful output for the **offline** transformer.
>
> > The "Attention is a Test-Time Regression" view [...] meaningfully employed.
>
> We agree with the reviewer that, while softmax attention performs the same mathematical operation as regression, this fact does not, on its own, imply that the framing is correct. In other words, it does not inherently prove that other key-value regression formulations will work well.
>
> Though, by building upon previous works [Wang, 2025], we show this generalization of attention is actually useful because natural improvements within this regression framework (adding weight decay [Yang, 2024b], momentum [Behrouz, 2024], global optimality [von Oswald, 2025], or more advanced optimizers like muon over SGD [Behrouz, 2025a]) has led to better sequence operation, regarding language modeling perplexity / downstream task performance. Additionally, our off-the-cuff example in Eq 6 predicted this concurrent work [6].
> Of course, you do not need this regression formulation to make a useful sequence operation (see LSTM). Likewise, not every instantiation of test-time regression will result in a good sequence model. We argue that this generalization of attention broadens the interpretability of these models.
>
> > Based on the above (mis)conception [...] misleading.
>
> > In the discussion of "Question 1", the paper concludes that "mapping past keys to values is not the end goal,  must map the query to the correct output ". Then the initial formalism was not correct in the first place?
>
> Like most of ML, while we fit to the training dataset, the end goal is to perform well on the test set. This does not mean that fitting to the training data is incorrect.
>
> > The examples present [...] actually exposes.
>
> In Eq 7, along with Eq 3 & 4, we state that parametric regressors *estimate the underlying key-to-value map* with parameters theta. During the online optimization, we fit this function so that $m_t(k_i|\theta_t) \approx v_i$.
>
> > predicting "values from keys" [...] cannot be a solid basis for life-long learning.
>
> For a lifelong agent to learn from experience and carry out long-term goals, the agent must create and solve its own subproblems [5]. These subproblems cannot be predefined before deployment, should limit human priors, and must remain general across domains.
>
> We suggest that transformers are already leveraging this fundamental building block for online learning with key-value estimation. For different contexts, the transformer constructs a online training dataset, providing a new regression subproblem to solve. This construction is flexible and learnable offline during pretraining.
>
> >  their limitations are also not very clear as ... limited representation of it).
>
> We will revise our language in 4.2 to note this.
>
>
> **Q1**:  "On a higher conceptual level, ..."
>
> If we lived in an infinite compute world, softmax attention would have some features of lifelong learning, but is known for its limited in expressivity (in TC0) compared to traditional RNNs (typically NC1), though this is out of the scope of this work.
>
> We believe that transformers offer a simple starting point to tackle lifelong learning through in-context learning. Our position is that extending in-context learning in transformers to lifelong settings requires a bounded representation of past context, when considering practical fixed hardware budgets.
>
> [5] https://arxiv.org/pdf/2208.11173
>
> [6] https://arxiv.org/pdf/2602.10410

---

> > ### Author Rebuttal · Reviewer_qDWj · 2026-04-02
> >
> > I thank the authors for clarifications.
> >
> > I have to acknowledge the point that viewing attention as an on-line regression or as a memory module is useful, in particular in the view of the mentioned works, which I am not familiar with.
> >
> > I am still struggling with the following:
> >
> > I am still confused by the concept of "regression", outside of the common understanding as in the statistical setup. When the (in-context) training data is a single highly dependent sequence (vs i.i.d. samples from a fixed distribution), and the objective is not specified (vs. expected loss on the same distribution), what is it in the end? The paper suggests that attention is a function $m_t$ that when queried with past keys would return past tokens and "interpolate" in between. But there is no statistically specified regression target. Still the paper says "$m_t$ generalizes" -- in what sense, if not the statistical one?
> >
> > In transformers, I believe a query may be completely different from all past keys, leading to a wider averaging of past values, effectively extracting different statistics. One example is when keys would be constant (assume) but with RoPE, then SA would perform a second order oscillating filter of values like in SSM (up to softmax normalization). During the training the transformer learns to explore and exploit all such possible ways of querying the "memory" $m_t$.  Does that still bear any analogy with a regression? I find that it could be misleading. On the other hand, do the authors see the evidence that such "filtering" accesses are unnecessary and only the focused differentiable memory lookups would suffice for Transformers?
> >
> > Let me clarify my prior example. The values in the transformer may be, without loss of generality, assumed equal to the keys. Namely, given a transformer with $(k,v,q)$ an equivalent transformer can be constructed with $k' = v' = (k,v)$ and $q' = (q, 0)$ and a suitable value projection, discarding $k$ part. Then the identity mapping $m_t(x) = x$ satisfies the perfect "regression" requirement (it maps all on-line training pairs correctly $k' \mapsto v'$ and even perfectly "generalizes" to the unseen key-value pairs), but obviously contains no information. If such (parametric) memory is substituted, the transformer obviously cannot make any use of it. Would any of the proposed regression or in-context learning views of key-> value mappings be credible in this setup? If not, why would it be credible if there is some ideological separation into keys and values but without a principled backing?
> >
> > In (7) should not the LHS and RHS be swapped?

---

### Official Review · Reviewer_SvhQ · 2026-03-18

**Significance:** 4
**Argument Clarity:** 4
**Rating:** 6
**Confidence:** 3

**Questions:**

I've included my questions earlier.

**Alternative Views Section:**

Yes

**Compliance With Llm Reviewing Policy A Conservative:**

Affirmed.

**Discussion Potential:**

4

**Final Justification:**

I keep my current scoe.

**Paper Summary:**

The paper states a position that in order to create agents with lifelong learning abilities we need to develop a different (parametric) types of attention. Authors claim that current generation of models is restricted to non-parametric type of attention (softmax-based) which grows with time (context length) making it harder (or even impossible) to create long enough contexts for the needs of the lifelong learning agents. The paper clearly presents its claims and rigorously explain justifications for each of the nuanced concepts. As a summary, instead of clearly outlining the path forward for the community, the work offers excellent research questions which stimulate the discussions around this topic.

**Position:**

Yes

**Position In Title:**

Yes

**Related Work:**

4

**Strengths And Weaknesses:**

Before going into the details of my review I'd like to thank the authors for their incredible work. It literally changed my perspective on the problem of attention mechanism and helped me develop a clear map of recent approaches, how they differ, what are the strengths and weaknesses of each. This is quite a rare situation so I wanted to highlight this fact at the beginning.

## Strengths
1. Topic & importance - I'd argue this is one the most important aspects of current development of modern architectures and the insights provided by the authors will be important for a broad part of community working on this topic.

2. Precision of the argumentation -- the arguments are balanced and clearly supported with a broad literature review leaving little to no room for any type of misunderstandings.

3. The open questions idea -- I find this approach to be way more effective than simply throwing some more or less justified statements about the future directions for the field.

## Weaknesses

1. The first reading of the paper was really hard for me. Clearly, I was missing the right framework to think about this work and it made the first read a hard task. The following were much better and gave me much better understanding of the overall position. I'm wondering whether you could somehow introduce the reader better to the correct way of thinking about this work. For instance, Figure 1 was not particularly helpful in my case, and I believe that a good illustration could help a lot in this case.

2. Your position states that: "extending in-context learning in transformers to lifelong settings requires parametric forms of attention". From my perspective, it implicitly assumes that in-context learning is the only way to enable lifelong learning for transformers and therefore we have to find out a way how to scale the context to arbitrary lengths. Is my interpretation correct? If so, I believe there are some alternative approaches that try to enable lifelong learning through weight updates (what was typically studied in continual learning community for years) or a hybrid approach that mixes both in-context learning and in-weights training such as [1]. I believe this should be somehow discussed as an alternative view for the problem, but I currently miss it. Please correct me if I misunderstood your position and my conclusions are wrong.

[1] https://arxiv.org/pdf/2602.15902

**Support:**

4

---

> ### Author Rebuttal · Authors · 2026-03-31
>
> > Before going into the details of my review I'd like to thank the authors for their incredible work. It literally changed my perspective on the problem of attention mechanism and helped me develop a clear map of recent approaches, how they differ, what are the strengths and weaknesses of each. This is quite a rare situation so I wanted to highlight this fact at the beginning.
>
> We really appreciate reading this. Thank you for your thorough review and giving this work more than just a first read.
>
> >The first reading of the paper was really hard for me. Clearly, I was missing the right framework to think about this work and it made the first read a hard task. The following were much better and gave me much better understanding of the overall position. I'm wondering whether you could somehow introduce the reader better to the correct way of thinking about this work.
>
> Clearly from your comment, as well as the other reviewers, our language in the abstract and introduction is not as sharp as it could have been. We will heavily revise our draft with this in mind.
>
>
> > For instance, Figure 1 was not particularly helpful in my case, and I believe that a good illustration could help a lot in this case.
>
> We will try to revamp this figure to improve the papers reception across audience backgrounds.
>
> >Your position states that: "extending in-context learning in transformers to lifelong settings requires parametric forms of attention". From my perspective, it implicitly assumes that in-context learning is the only way to enable lifelong learning for transformers and therefore we have to find out a way how to scale the context to arbitrary lengths. Is my interpretation correct?
>
> This is pretty much correct, though in-context learning is not necessarily the only way to enable lifelong learning for transformers. In our introduction, we say that in-context learning with transformers is a straightforward starting point for lifelong learning. There are numerous paths to lifelong learning and we cannot say with certainty that others won't work. We believe that extending in-context learning is a simple path with tangible checkpoints, given that both research and industry have cemented transformers as the default model. While lifelong learning will require some form of paradigm shift, extending in-context learning seems to be a smaller “shift” than other options like traditional re-training with continual learning.
>
> Then, in attempt to extend in-context learning in transformers to lifelong contexts, our position is that this requires a bounded representation of past context (i.e. parametric regressors in our framework).
>
> > If so, I believe there are some alternative approaches that try to enable lifelong learning through weight updates (what was typically studied in continual learning community for years) or a hybrid approach that mixes both in-context learning and in-weights training such as [1]. I believe this should be somehow discussed as an alternative view for the problem, but I currently miss it. Please correct me if I misunderstood your position and my conclusions are wrong.
>
> Yes, we will include this work and others, like “TTT-E2E” [7], that perform test-time training to the whole model weights, beyond attention’s “online parameters” as another alternative view.
>
> We will update our introduction to better discuss this, and include another alternative view to show that extending in-context learning is not necessarily the only solution to lifelong learning.
>
> [7] https://arxiv.org/abs/2512.23675

---

> > ### Author Rebuttal · Reviewer_SvhQ · 2026-04-07
> >
> > I only had some minor comments which were fully resolved during the rebuttal. I keep my score strongly voting for accepting the work.

---

### Decision · Program_Chairs · 2026-04-30

**Decision:**

Reject

**Comment:**

Overall, the paper presents an ambitious and timely position paper on the role of attention mechanisms to enable lifelong learning. Reviewers agree that the problem is important and the paper successfully unifies diverse approaches to attention. However, reviewer qDWj has serious concerns about the rigor of the paper, specifically related to the analogy to regression. There are also additional concerns about the lack of empirical validation, clarity of the exposition and strength of certain claims. Overall, there are clear contributions to the paper but concerns as to the clarity and some disagreement on viewpoint.